# Strongyloidiasis Serological Analysis with Three Different Biological Probes and Their Electrochemical Responses in a Screen-Printed Gold Electrode

**DOI:** 10.3390/s21061931

**Published:** 2021-03-10

**Authors:** Francielli C. C. Melo, Luciano P. Rodrigues, Nágilla D. Feliciano, Julia M. Costa-Cruz, Vanessa S. Ribeiro, Bruna F. Matias-Colombo, Renata P. Alves-Balvedi, Luiz R. Goulart

**Affiliations:** 1National Agency for Health Surveillance-Brasília, SIA Trecho 5, Área Especial 57, Bloco A/B, 1° Andar, Brasília, DF 71205-050, Brazil; francielli.melo@anvisa.gov.br; 2Institute of Engineering, Science and Technology, Federal University of the Jequitinhonha and Vale de Mucuri, Av. Um, n. 4.050—Cidade Universitária, Janaúba, MG 39447-790, Brazil; luciano.rodrigues@ufvjm.edu.br; 3Laboratory of Parasite Diagnosis, Institute of Biomedical Sciences, Federal University of Uberlandia, Av. Amazonas s/n Bl. 4C, sl. 239, Uberlândia, MG 38400-902, Brazil; nagilla@iftm.edu.br (N.D.F.); costacruz@ufu.br (J.M.C.-C.); vanessa.ribeiro@ufu.br (V.S.R.); 4Laboratory of Nanobiotechnology, Institute of Biotechnology, Federal University of Uberlandia, Av. Amazonas s/n Bl. 2E, sl. 248, Uberlândia, MG 38402-022, Brazil; brunafrancamatias@yahoo.com.br (B.F.M.-C.); lrgoulart@ufu.br (L.R.G.); 5Biological Science, Federal University of Triângulo Mineiro, Rua Antônio Baiano, n 150, Iturama, MG 38280-000, Brazil

**Keywords:** electrochemical immunosensors, *Strongyloides stercoralis*, single-epitope, synthetic peptides, multiple-epitope, human IgG

## Abstract

(1) Background: The validation of biological antigens is the study’s utmost goal in biomedical applications. We evaluated three different probes with single and multiple epitopes through electrochemical detection of specific IgG in serum for human strongyloidiasis diagnosis. (2) Methods: Screen-printed gold electrodes were used and probes consisting of two single-epitope synthetic peptides (D3 and C10) with different sequences, and a multi-epitope antigen [detergent phase (DP)—hydrophobic membrane proteins]. Human serum samples from three populations were used: *Strongyloides stercoralis* positive, positive for other parasitic infections and negative controls. To test the immobilization of probes onto a screen-printed gold electrode and the serum IgG detection, electrochemical analyses were carried out through differential pulse voltammetry (DPV) and the electrode surface analyses were recorded using atomic force microscopy. (3) Results: The electrochemical response in screen-printed gold electrodes of peptides D3 and C10 when using positive serum was significantly higher than that when using the DP. Our sensor improved sensitivity to detect strongyloidiasis. (4) Conclusions: Probes’ sequences are critical factors for differential electrochemical responses, and the D3 peptide presented the best electrochemical performance for strongyloidiasis detection, and may efficiently substitute whole antigen extracts from parasites for strongyloidiasis diagnosis in electrochemical immunosensors.

## 1. Introduction

Strongyloidiasis is a worldwide neglected public health disease caused by *Strongyloides stercoralis* [1,2], and although it may result in long-lasting infection and death, epidemiological data are relatively scarce. Approximately 370 million people are estimated to be infected worldwide, and Brazil is one of the countries that have the most extensive studies on the prevalence of strongyloidiasis, with reported prevalence of 13%. Standard diagnosis of strongyloidiasis relies on the demonstration of the parasite, but culture methods and polymerase chain reaction have also been used [3]. Serological assays, such as enzyme linked immunosorbent assay (ELISA), are often used for antibody detection, showing different sensitivity and specificity values depending on the antigens used for detection.

After several studies, new antigens have been used in serological analyses (ELISA) to detect IgG in biological samples. Three probes showed great diagnostic potential due to their high sensitivity and specificity: a detergent phase (DP)—multi-epitope proteins extracted from *S. venezuelensis*, formed by a pool of hydrophobic proteins of the parasite [4], and two mono-epitope peptides with different amino acid sequences, named C10 and D3, which consisted of two repetitive epitope motifs spaced by GGGS with some structural modifications [5].

Quantification of biological constituents through electrochemical immunosensors requires conversion of biological information into electronic signals; however, electrodes do not always work with specific probes, probably due to the complexity of the biological environment. Connection of electronic devices with functionalized electrodes is dependent on different materials. We believe that the direct conversion of a biological event into an electronic signal relies on the interaction of materials’ interfaces with biological samples, which cannot be seen in a simplistic view.

The understanding of fluids’ behavior at the interfacial layer next to the surface of a solid material is critical for sensing applications [6]. When the solid surface is charged, it can drive further changes in the interfacial liquid. This phenomenon has been explored through the characterization of the interfacial water structure at the surface of bare gold electrodes. With positive potentials or no potential, the layer was highly structured, and at negative potentials, the layer was more like less dense water [7].

Therefore, it is expected that complex fluids and probes will present multifaceted performances in different potentials, and therefore, sensors must be extensively optimized, and probably conventional parameters of immunoassays cannot be reproduced in electrochemical sensors.

Gold electrodes have important characteristics that make them ideally suited for electrochemical immunosensing, which include large surface area, superior conductivity, biocompatibility, high stability, and a well-characterized thiol–gold surface chemistry. The high surface area enables a greater amount of probes to be immobilized onto the surface relative to that obtained on a flat, planar surface, providing greater access to the target substrate [8,9].

Given the above, we have developed a simple diagnostic test for strongyloidiasis IgG detection in sera, using three different antigens as probes. ELISA standards for each antigen were compared with the device responses. We hypothesized that interactions of biological probes with gold electrodes are dependent on probes’ sequences, amount and diversity, which may lead to differential electrochemical responses, and such aspects will be explored herein.

## 2. Materials and Methods

### 2.1. Ethics Statement

This study was conducted according to the rules of the declaration of Helsinki of 1975 and the ethical guidelines of the Brazilian Ministry of Health, with protocols and procedures approved by the Research Ethics Committee/UFU, under the protocol number 307.605, approved in 2013. The serum samples used in this study were stored in the Biological Samples Bank of the Parasitic Diagnosis Laboratory of the Federal University of Uberlandia, under approval by the Research Ethics Committee/UFU, under protocol number 041, approved in 2008.

### 2.2. Study Population

This study used serum samples divided into three groups of patients. Group 1 (G1) consisted of pooled samples (*n* = 3) from patients living in an endemic area with confirmed parasitological diagnosis of strongyloidiasis [10,11], based on positive larval thermo-hydrotropism and the method of a gravity sedimentation technique [12]. Group 2 (G2) included pooled samples from patients with positive diagnosis of other parasitic diseases, including: *Ascaris lumbricoides* (*n* = 8), *Enterobius vermicularis* (*n* = 5), *Giardia lamblia* (*n* = 5), hookworm (*n* = 7), *Hymenolepis nana* (*n* = 4), *Schistosoma mansoni* (*n* = 4), *Taenia* sp. (*n* = 6), and *Trichuris trichiura* (*n* = 5).

Samples from Groups 1 and 2 were obtained from patients affected by single infection. Group 3 (G3) contained pooled samples from apparently healthy individuals (*n* = 3), based on clinical observation without any evidence of contact with *S. stercoralis* infection, and with three negative tests of fecal samples.

### 2.3. Probes

Antigens used in this study were chosen based on previous diagnostic tests obtained in serological tests (ELISA), which detected IgG in serum samples from patients with strongyloidiasis according to the conditions presented in Table 1.

The three antigens consisted of two mono-epitope peptides with different amino acid sequences, named C10 and D3 and detergent phase (DP) [4,5].

The isoelectric points of peptides were calculated through the server “isoelectric.org” [13], to aid the design of the study regarding the previous steps of preparation of the electrodes before the functionalization of the biomolecules.

### 2.4. Electrodes and the Potentiostat Apparatus

All electrochemical assays were conducted in triplicate, at room temperature (25 ± 1 °C), performed in a low temperature-synthesized gold screen-printed electrode (BT 220, DropSens, Asturias, Spain, working and auxiliary electrodes are made of gold, while reference electrode is available in silver) using the potentiostat PalmSens 3 (PalmSens, Houten, The Netherlands) with the PSTrace 5.4 software (PalmSens, Netherlands).

### 2.5. Electrochemical Measurements

To assess the conductivity of electrodes, cyclic voltammetry (CV) was carried out in the gold screen-printed electrode, without any biomolecule, and aqueous indicator electrolyte solution, then current changes were recorded.

Using the differential pulse voltammetry (DPV) technique, we evaluated the behavior of the electrochemical signal of the supporting electrolyte (indirect detection) on the sensor to test the immobilization of probes onto the gold screen-printed electrode and the serum IgG detection.

The measurements for detections were carried out in 5 mM [Fe (CN)_6_]^−3/−4^ and 0.1 M KCl, pH 7.4, for the interaction investigation between total probe immobilized on working electrode and total soluble IgG recognition. Thus, 80 uL of this solution was pipetted over the three electrodes, closing the working electrode circuit between the other two electrodes (counter electrode and reference) [14].

### 2.6. Biosensor

Briefly, 2 µL of a solution with each probe: C10 or D3 (1.0 mg mL^−1^) or DP (5.0 mg mL^−1^) was added onto the working electrode surface of a gold screen-printed electrode (BT 220). The electrodes with probe were incubated for 20 min, washed with 1 mL phosphate buffer solution (0.1 mol L^−1^, pH 7.4) and dried. The working electrode surface was blocked with 2.0 µL of a BSA 0.5%, diluted in phosphate buffer solution, for 20 min to eliminate the non-specific binding effect and block the remaining active sites. Washing was performed again with 1 mL of phosphate buffer solution (0.1 mol L^−1^, pH 7.4) and electrodes were dried. For direct measurements, electrodes were incubated with 2 µL of pooled serum samples (dilution 1:160 for peptides D10 and C3 or 1:80 for DP) from patients from G1, G2 or G3, according to previous standardization in ELISA protocols summarized in Table 1, and incubated for 20 min, and then electrodes were washed again with 1 mL of phosphate buffer solution and dried as shown in Figure 1.

### 2.7. Electrodes’ Surface Analyses

The electrode surface analyses were recorded using Atomic Force Microscope (AFM) Model XE-70 manufactured by Park System^®^ and Scanning Electron Microscopy (SEM) 9600^®^ (Shimadzu Corp, Kyoto, Japan). The images were generated at the Laboratory of New Insulating and Semiconductor Materials and Multiuser Microscopy Lab of the Federal University of Uberlandia, MG, Brazil.

## 3. Results

### 3.1. Electrochemical Approaches

Electrochemical approaches with cyclic voltammogram and differential pulse voltammograms were obtained using the probes: C10 and D3 peptides, and detergent phase (DP), and the pooled serum samples as shown in Figure 2.

### 3.2. Probe Target–Working Electrode Surface Interaction

I-TASSER (https://zhanglab.ccmb.med.umich.edu/I-TASSER/output/S566621 accessed on 20 August 2020); simulations for C10 (ACAPSFFHSCGGGSACAPSFFHSC) and D3 (ACSLASPSLCGGGSACSLASPSLC) peptides show the structural conformations obtained through the SPICKER (https://zhanglab.ccmb.med.umich.edu/I-TASSER/output/S566845), accessed on 20 August 2020.

The confidence of each model is quantitatively measured by C-score that is calculated based on the significance of threading template alignments and the convergence parameters of the structure assembly simulations. Both structures shown have the largest C-score among all simulations performed (Figure 3).

### 3.3. Surface Analysis

Atomic force microscopy (AFM) was performed to evaluate the morphological changes on the surface of the electrode after the interaction of the peptide with the serum from Groups 1, 2 and 3. This technique is used to visualize images of surfaces with antigens (Ag) or antibodies (Ab) adsorbed or combined due to Ag-Ab recognition [15,16,17].

AFM was made through sweeping in 10 × 10 nm using the non-contact mode, and resolution of 10 nm. Structural and topographical observations of the sensors’ surfaces were performed prior and after the probe immobilization, after blocking solution, and after serum incubation.

Topographic images recorded roughness differences according to the degree of interaction/immobilization of the probe recognized by IgG on the gold surface (Figure 4).

The roughness coefficient (Rq) can be found in Table 2.

## 4. Discussion

This investigation presents for the first time a biosensing platform for strongyloidiasis detection, and also demonstrates how biological events of antigen-antibody complexes are transduced into electronic signals by showing that specific interactions of gold electrodes with probes are modulated by their epitope sequences and antigen complexity (number of epitopes). We have demonstrated that single epitope peptides are better suited for detection than protein fractions with multiple epitopes are.

It is important to note that standard diagnosis of strongyloidiasis still relies on the demonstration of the parasite on feces, although polymerase chain reaction has also been used [3]. On the other hand, serological analyses lack good diagnostic parameters, and their clinical use is not well established. However, we have recently developed specific peptides (C10 and D3) that presented excellent diagnostic parameters, with sensitivity of 95% and specificities of 89.2% and 92.5%, respectively [5]. However, previously another study from our group has also shown that protein fractions (detergent phase) showed sensitivity and specificity of 88.0% and 84.4% in serum, respectively [4]. These three probes were chosen to be compared based on their sequence and antigen complexity.

Firstly, we must consider that the detergent phase (DP) extracted from *S. venezuelensis* is formed by a pool of hydrophobic proteins of the parasite [4], and its interaction with the gold electrode may occur through hydrophobic, ionic interactions or chemical bonds by thiol groups, depending on the type and amino acid sequences that form these proteins. Due to the protein diversity and probably due to the predominance of weak physical interactions between gold and these amino acids, the surface functionalization will occur in a more disordered way and we will not always have the free epitopes for regions of free contact with the immunoglobulins [4,5,17,18]. In these cases, some of the probes could not recognize their target, which could interfere with the values of the oxidation current measured, explaining the smaller differentiation between the groups in the DP fraction.

In contrast, the C10 and D3 probes structures had a better interaction with the working electrode material because they are smaller and epitope specific. In a 2014 study by Palafox-Hernandez [15] about interactions between peptides and silver or gold surfaces, it has been shown that cysteine (Cys), arginine (Arg), tyrosine (Tyr) and tryptophan (Trp) have higher affinity for gold surfaces, and that the adsorption occurs essentially through direct contact between molecules, and not mediated by the solvent. The cysteine is covalently bonded to gold by dissociation of -SH group, forming S-Au bridges. Both synthetic peptides, C10 and D3, have four cysteine residues distributed almost equidistant from their 24 amino acids (Figure 3). The presence of cysteine in both probes (Figure 3) corroborates previous studies [19,20,21] that have shown the adsorption of cysteine on gold electrode surfaces through the formation of cysteine dimers, requiring no other mechanisms to functionalize the working electrode.

In addition to the peptide–gold interactions, we must consider that amino acids between distinct peptides form hydrogen bonds between each other, and these bonds lead to the formation of aggregates of peptides on the gold surface [22], increasing the amount of probe molecules on electrode surface and consequently the recognition of the target, as is verified in the oxidation potentials of the studied groups. This explains the difference between the results obtained between the peptides and detergent phase.

The reduction in the current values for G1, positive for strongyloidiasis, is probably due to the electron transfer kinetics block of ([Fe(CN)_6_]^−3/−4^ [23,24]. This occurs because both the peptide probe and target IgG are not electron-donating species for the system, generating resistivity between indicator electrolyte and gold surface.

Some amino acids are electroactive compounds, and many have known antioxidant properties in metabolic chains. Masek [25] describes that compounds with antioxidant properties have a low oxidation potential, like cysteine (half-wave potential 0.38 V) and its oxidation product cystine (half-wave potential 0.57 V), both with low electron-donating capacity in platinum electrode. Immunoglobulins tend to present positive charges at neutral pH, which can be seen in the electrophoresis by their migration in gamma globulin fraction. Therefore, these species would be important interferents in the flow of electrons of the system when combined to the antigens immobilized in the electrode surface. The combination of all these factors possibly led to the reduction in the oxidation peak values in the readings of positive serum group, which did not occur for other parasitic diseases and negative control samples, due to the absence of the anti-*Strongyloides* IgG in these sera.

Besides, the screen-printed gold electrode is responsible for increasing the Faradaic current due to the larger area of the electrode in contact with the reagents, consequently increasing its sensitivity and limits of detection [25]. Our system was able to detect anti-*Strongyloides* IgGs in small serum fractions of 2 microliters at the dilutions of 1:80 or 1:160, according to the probe used.

Although the evaluated serum contains many interfering substances, such as urea, uric acid, glutamate, and albumin, we can consider that there was no disturbance of the detection process by these substances [26]. The best results using C10 and D3 for the detection of strongyloidiasis agree with previously published reports [7,26], demonstrating that the peptides obtained from Phage Display selections are more specific than the pool of membrane peptides, such as DP.

Our findings also demonstrate that the D3 peptide was superior in electrochemical response probably due to its higher specificity when compared to C10, which is equivalent to saying that the higher the specificity, the higher the antibody affinity. Another possibility is that the D3 probe may bind more strongly to the gold electrode since it has two more sulfur atoms in its sequence (Figure 3). In addition, in the D10 probe, sulfur-containing amino acids are adjacent to bulky aromatic rings that can harm S-Au binding. The peptides synthesized chemically proved to be an important and versatile diagnostic tool to detect the target IgG [7,15,27].

The AFM surface analysis (Figure 4 and Table 2) showed that the D3 probe presented increased roughness on seropositive group (G1) electrodes, and decreased Rq after the addition of positive serum when compared to other parasites samples (G2) and negative controls (G3). This result corroborates the voltammetry graphs in which the DP and C10 present little differentiation among curves; while the D3 peptide exhibits the greatest differentiation among G1, G2 and G3 groups. Thus, roughness increases when positive serum antibodies are recognized with greater efficiency [28].

The difference in the roughness coefficient found in our experiments can be explained by the fact that the topography of adsorption and the interaction between biomolecules involved were determined by the composition, amino acid sequence of the peptide, and affinity to the antibodies. These characteristics determine the position of the binding site within the peptide structure and the physicochemical interactions between peptide, the electrode surface and between peptides themselves (hydrophobic interactions, hydrogen bonds, covalent bonds, electrostatic interactions) with the additional influence of environmental conditions, such as pH and ionic strength [16,17,28].

Therefore, both AFM images and electrochemical analyses not only confirmed the successful immobilization protocol, but also the greatest specificity and sensitivity of the D3 peptide.

Briefly, our results indicate that the direct conversion of a biological event into an electronic signal relies on the specific interaction of nanomaterials’ interfaces with probes. It is important to note that the mimotopes selected by Phage Display and its synthetic version used in this study, represent highly reactive and specific epitopes of proteins, which under proper design and together with gold electrodes, may lead us to build stable electrochemical platforms for disease diagnostics.

## 5. Conclusions

This work shows the validation of biological antigens through electrochemical detection of specific IgG in serum for human strongyloidiasis diagnosis. We have demonstrated that probes’ sequences are critical factors for differential electrochemical responses. Besides, the D3 peptide presented the best electrochemical performance for strongyloidiasis detection and may efficiently substitute whole antigen extracts from parasites for strongyloidiasis diagnosis in electrochemical immunosensors.

## Figures and Tables

**Figure 1 sensors-21-01931-f001:**
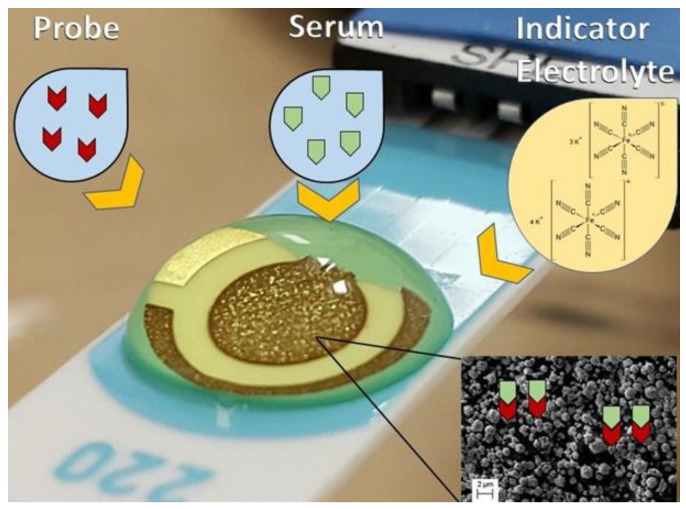
Schematic illustration of the experimental procedure performed. Probe was immobilized at the surface of the working electrode. An indicator electrolyte a solution containing [Fe(CN)6]^−3/−4^ 5.0 mM and KCl 0.1 M was used, V = 100 mV s^−1^. Inset shows the micrometric view of gold at the surface of the electrode.

**Figure 2 sensors-21-01931-f002:**
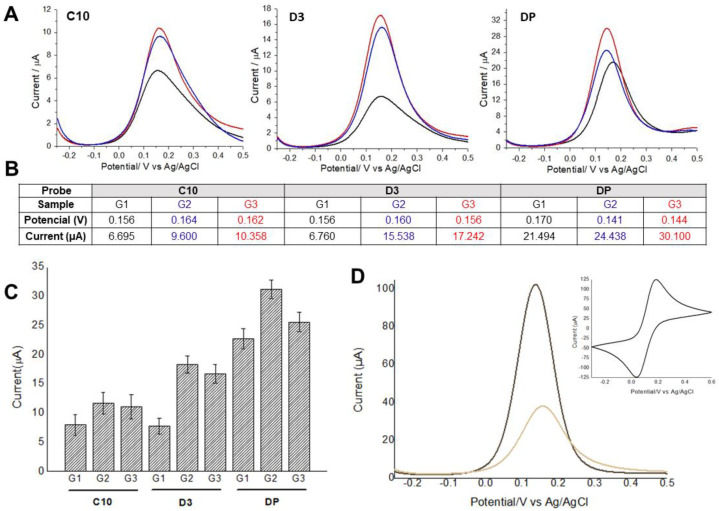
(**A**) Differential pulse voltammograms obtained using C1 and D3 peptides, and detergent phase (DP) with pooled serum samples from Groups 1, 2, and 3. Black lines indicate *Strongyloides*-positive samples (G1), blue and red lines represent samples from patients with other parasitic diseases (G2) and negative controls (G3), respectively. (**B**) Average values of potential and oxidation peak current for each voltammogram. (**C**) Average values of potential and oxidation peak current and their respective standard deviations (*n* = 3). (**D**) Differential pulse voltammograms analyses were performed before and after probe immobilization, followed by BSA application as blocking agent to prevent unspecific binding of antibodies. The insert shows the cyclic voltammogram used to assess the electrodes’ conductivity. Indicator electrolyte solution [Fe(CN)_6_]^−3/−4^ and KCl, electrode BT 220, scan rate = 15.0 mV s^−1^.

**Figure 3 sensors-21-01931-f003:**
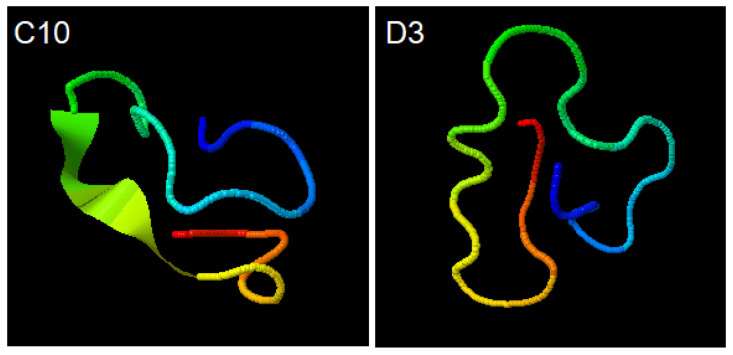
Simulations for C10 (ACAPSFFHSCGGGSACAPSFFHSC) and D3 (ACSLASPSLCGGGSACSLASPSLC) peptides. Structural conformations obtained through the SPICKER.

**Figure 4 sensors-21-01931-f004:**
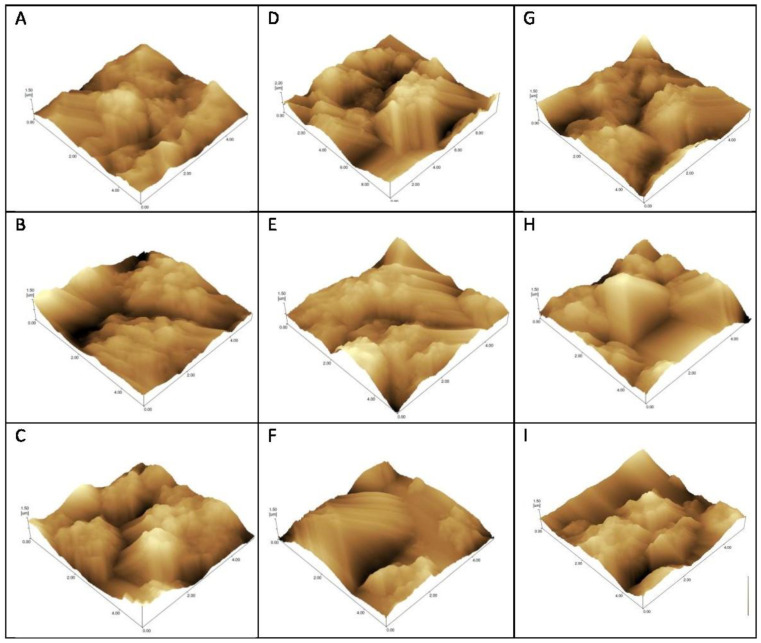
Atomic force microscopy topographical images of gold electrodes for detection of IgG using C10 (**A**–**C**), and D3 (**D**–**F**) peptides and detergent phase (**G**–**I**) as probes. Incubation with *Strongyloides*-positive samples—G1 (**A**,**D**,**G**), samples from patients with other parasitic diseases—G2 (**C**,**F**,**I**), and negative controls—G3 (**B**,**E**,**H**).

**Table 1 sensors-21-01931-t001:** Three probes used for ELISA protocols.

Antigen (Ag)	Reaction Condition	Reference
Peptide C10 ^1^	1 mg/mL (Ag/well); Serum 1:160	[5]
Peptide D3 ^1^	1 mg/mL (Ag/well); Serum 1:160	[5]
Detergent Phase (DP)	5 mg/mL (Ag/well); Serum 1:80	[4]

^1^ The structural conformations of each probe were obtained through the SPICKER.

**Table 2 sensors-21-01931-t002:** Roughness coefficient (Rq) for antigen binding to patients’ serum.

Antigen (Ag)	Bare Gold	G1	G2	G3
Peptide C10	230	187.06 ± 1.22 a	274.96 ± 71.51 a	192.28 ± 12.35 a
Peptide D3	250	651.42 ± 13.58 a	214.85 ± 26.81 b	353.43 ± 3.81 b
Detergent Phase (DP)	220	233.87 ± 22.29 a	275.42 ± 5.26 a	303.06 ± 23.09 a

Rq means and their standard errors followed by different letters in the same column are statistically significant (*p* < 0.05).

## Data Availability

Not applicable.

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
