# Peer review of "Strongyloidiasis Serological Analysis with Three Different Biological Probes and Their Electrochemical Responses in a Screen-Printed Gold Electrode"

_sensors, 2021, doi:10.3390/s21061931_

Round 1
Reviewer 1 Report
In the work by Melo et al., the authors characterize the performances of electrochemical biosensors for markers of strongyloidiasis in diluted serum. The authors compare the results from three sensing units in terms of electrical signal and roughness of the electrode surface.
The work is interesting since it validates a standard strategy for biosensing, but the manuscript is a bit messy.
Few suggestions to make it clearer to the reader:
- The abstract should be rephrased.
- The introduction section about Strongyloidiasis should be deepened; in particular I recommend a better comparison with the detection state-of-the-art.
- Since the authors are using nanoporous gold electrodes and BSA as blocking agent, a study about coverage should be added. A simple cyclic voltammetry analysis before and after immobilization of the sensing unit should be sufficient.
- The authors state that “Considering that IgG is much larger than the probes, a rougher surface is expected on seropositive group electrodes”, but for DP and C10 sensors, positive serum has the minimum Please comment.
- In table 2, the authors report several time letters “a” and “b” after the values, but they do not describe the reason why.
Author Response
Thank you very much for handling our manuscript.
After carefully revising the entire manuscript, including all suggestions and considerations of the reviewers’ comments, we submit the revised manuscript, entitled “Strongyloidiasis serological analyses with three different biological probes and their electrochemical responses in screen-printed gold electrode” (Manuscript ID: sensors-1115385). The revised version with specific changes are highlighted, and point-by-point response is presented to each comment raised by the reviewers.

Reviewer 2 Report
Dear Authors,
in my opinion your draft needs several major improvements before considering the publication in this respected Journal. Please consider the following suggestions for future versions of this draft.
1) This draft needs a deep and careful revision of the English form, as well as of the data presentation. In this version, you are omitting several critic steps both in the functionalization process and in the data analysis dissertation, and this choice hinders the full understanding of your activity.
2) Please briefly describe the state-of-the-art of the sensing platforms for this target.
3) (Introduction) please clarify what is the connection between AFP sensor and Strongyloidiasis one, and why the former is propaedeutic to the latter.
4) Par. 2.4 I often use DropSens sensors, but I can hardly define them as “nanoporous”, as they are fabricated through screen printing techniques. Please justify your definition.
5) Par. 2.5 Please declare the nature of your indicator electrolyte solution and its purpose in your preliminary characterization measurements.
6) Par. 2.5 please clarify the materials that constitute both worker, reference and counter electrodes of your electrochemical set up.
7) Par. 2.6 Please clarify why you did not use any blocking molecules at the electrode surface to avoid unspecific binding, e.g., MCH, BSA, etc.
8) Par. 3 I suggest that you add the preliminary characterization of your sensor with CV to prove your initial assumptions.
9) Please discuss why you did not use tools like electrochemical impedance spectroscopy to assess the outcome of the functionalization process, rather than using only AFM data.
10) please clarify the meaning of rr. 170-171. Also, please discuss in greater detail the implication of such sentence.
11) Fig. 2 please clarify the meaning of PSS, NSS, OPD.
12) please address the following typos and/or rephrase the following sentences to improve text readability:
-) rr. 25-26 “… were considered parameters to verify and compare whether specific biological factors… “
-) rr. 31-32 “Our results bring breadth to sensitivity of electro-chemical immunosensors”
-) rr. 33-34 “… probably conditioned a greater quantity of a unique epitope”
-) r. 123 “… and to optimization the detection… “
-) r. 124 “… were carried out the nano porous gold… “
-) please rephrase rr. 132-134 from “Briefly…” to “… electrode surface”
-) Par. 3.2 please substitute “work electrode” with “working electrode”
-) please rephrase the whole r. 226
Kind regards.
Author Response

(The authors gave the same response as above.)

Reviewer 3 Report
The manuscript entitled The strongyloidiasis serological analysis with three different biological probes and their electrochemical responses in screen-printed gold electrode submitted by the group of Authors represents and interesting research on potential sensitivity enhancement and identify what antigen shows higher specificity due to the highest and most specific signal.
The work is interesting and easy to follow.
Comments:
Abstract should be focused on what the research was about. Please be more punctual in description of results and conclusions.
Conclusion should be arranged in a way the reader could easily follow the main conclusion of the research. Conclusion should be linked with the Abstract.
In general, very interesting work.
Author Response

(The authors gave the same response as above.)

Round 2
Reviewer 2 Report
Dear Authors,
thank you for your fuitful redrafting efforts. Please modify "analyses" with "analysis" in the title.
Kind regards.
Author Response
Reply to the Reviewer 2 (Minor revisions)
Comments to the Reviewer 2:
- We are now submitting the revised manuscript with highlighted changes together with a point-by-point response below.
- We have significantly improved the introduction, the methods and conclusions were rewritten according to reviewers’ comments and results are clearly presented, with English language and spell check.
- We are very grateful for your efforts to help us make this paper better and inform you that all corrections were made in an attempt to correctly address the points suggested by the reviewers.
Comments and Suggestions of Reviewer 2 and answers of Authors:
Dear Authors, thank you for your fruitful redrafting efforts. Please modify "analyses" with "analysis" in the title.
We would like to thank you again for your efforts to help us and inform you that we changed analyses with analysis in the title (highlighted green).